# Glass-Ceramic Fillers Based on Zinc Oxide–Silica Systems for Dental Composite Resins: Effect on Mechanical Properties

**DOI:** 10.3390/ma16186268

**Published:** 2023-09-19

**Authors:** Peyman Torkian, SayedMohsen Mortazavi Najafabadi, Majid Ghashang, Dariusz Grzelczyk

**Affiliations:** 1Department of Manufacturing, Faculty of Mechanical Engineering, Babol Noshirvani University of Technology, Babol 47148-71167, Iran; 2Department of Automation, Biomechanics and Mechatronics, Lodz University of Technology, 1/15 Stefanowski Street, 90-537 Lodz, Poland; 3Department of Chemistry, Najafabad Branch, Islamic Azad University, Najafabad 15847-43311, Iran

**Keywords:** Zn_2_SiO_4_, SiO_2_–ZnO, glass ceramics, mechanical properties, dental composites, flexural strength, flexural modulus, compressive strength

## Abstract

The potential of glass ceramics as applicable materials in various fields including fillers for dental restorations is our guide to present a new procedure for improvements of the mechanical properties of dental composites. This work aims to use Zn_2_SiO_4_ and SiO_2_–ZnO nano-materials as fillers to improve the mechanical properties of Bis-GMA/TEGDMA mixed dental resins. Zn_2_SiO_4_ and SiO_2_–ZnO samples were prepared and characterized by using XRD, FE-SEM, EDX, and FT-IR techniques. The XRD pattern of the SiO_2_–ZnO sample shows that ZnO crystallized in a hexagonal phase, while the SiO_2_ phase was amorphous. Similarly, the Zn_2_SiO_4_ sample crystallized in a rhombohedral crystal system. The prepared samples were used as fillers for the improvement of the mechanical properties of Bis-GMA/TEGDMA mixed dental resins. Five samples of dental composites composed of Bis-GMA/TEGDMA mixed resins were filled with 2, 5, 8, 10, and 15 wt% of SiO_2_–ZnO, and similarly, five samples were filled with Zn_2_SiO_4_ samples (2, 5, 8, 10, and 15 wt%). All of the 10 samples (**A_1_**–**A_10_**) were characterized by using different techniques including FT-IR, FE-SEM, EDX, and TGA analyses. According to the TGA analysis, all samples were thermally stable up to 200 °C, and the thermal stability increased with the filler percent. Next, the mechanical properties of the samples including the flexural strength (FS), flexural modulus (FM), diameter tensile strength (DTS), and compressive strength (CS) were investigated. The obtained results revealed that the samples filled with 8 wt% of SiO_2_–ZnO and 10 wt% of Zn_2_SiO_4_ had higher FS values of 123.4 and 136.6 MPa, respectively. Moreover, 8 wt% of both fillers displayed higher values of the FM, DTS, and CS parameters. These values were 8.6 GPa, 34.2 MPa, and 183.8 MPa for SiO_2_–ZnO and 11.3 GPa, 41.2 MPa, and 190.5 MPa for the Zn_2_SiO_4_ filler. Inexpensive silica-based materials enhance polymeric mechanics. Silica–metal oxide nanocomposites improve dental composite properties effectively.

## 1. Introduction

For years, scientists’ efforts have been focused on providing polymer composites with improved chemical and mechanical properties for applications such as the construction and restoration of dentals. These activities have resulted in products with improved resistance and strength. Polymer composites used in dentistry include three different components: The first component includes organic monomers that form the polymer mixture and usually form the main component of the composite. The second component is the radical initiators, which include a small percentage of the mixture. Finally, the third includes fillers that are usually used to increase the mechanical strength of dental composites [1]. Among the used monomers, bisphenylglycidyl dimethacrylate (Bis-GMA) and triethyleneglycol dimethacrylate (TEGDMA) have been widely used due to their easy polymerization, the high chemical stability of the produced polymers, and finally, their suitable waterproof properties. On the other hand, the main role of fillers is to improve the mechanical properties of the composite, including its tensile and bending strength, and silica-based fillers have been used the most [1].

The development of fillers can be considered the main research field of researchers to provide a method to improve the mechanical properties of dental composites. Revisiting the research conducted in the field of optimizing the properties of dental polymers, there are many interesting reports. In other words, many fillers have been used to improve the properties of dental polymers, some of which are polyacrylonitrile (PAN) nanofibers [2], carbon nanotubes modified by silane coupling agents [3], Ag nanoparticles [4], quaternary ammonium polyethylenimine nanoparticles (QA-PEI-NPs) [5], bioactive glass [6], SiO_2_ nanofibers [7], alumina particles [8], spherical silica [9], hydroxyapatite whisker [10], nano fibrillar silicate [11], nano-zirconia fillers [12], electrospun nano-scaled glass fiber [13], NaF-loaded core–shell PAN–PMMA nanofibers [14], and APTES- or MPTS-conditioned nanozirconia [15].

In addition, glass ceramics have found many applications in various industries due to their low prices, easy production methods especially on a commercial scale, and high crystallization ability. Notably, glass ceramics composed of ZnO and SiO_2_ phases have suitable potentials as luminescence materials to be used in various optoelectronic devices; thermal, chemical, and mechanical stability; and excellent transparency for glass phosphorous materials. Glass ceramics are suitable candidates for use as fillers in dental composites due to their high transparency and sufficient strength and biocompatibility with the tooth. So far, many cases of glass ceramics have been used as fillers in dental composites. However, efforts to develop dental fillings based on glass ceramics have never ended [16,17,18].

The aim of this work is to develop two new systems to be used as a filler for dental composite reinforcements. In continuation of our studies on the preparation and application of nanostructured materials [19,20,21,22,23], here Zn_2_SiO_4_ and SiO_2_–ZnO nano-materials were used as resistant reinforcing particles in a dental composite to measure mechanical properties such as flexural strength, etc., after construction.

## 2. Materials and Methods

### 2.1. Reagents and Instrumentation

The salts, chemicals, and solvents used for this project were purchased from a commercial seller of Merck Chemicals (Merck Company, Darmstadt, Germany). The Bis-GMA, TEGDMA, CQ, and 3-NDADM were purchased from a commercial seller of Aldrich chemicals (Sigma-Aldrich, St. Louis, MO, USA). The thermal behavior was investigated to confirm the thermal stability of the samples using TGA analysis on a Shimadzu TGA-DTG-60H instrument (Kagoshima, Japan). The FT-IR analysis was performed on a Bruker spectrophotometer (BRUKER TEMSOR27, Borken, Germany) using the KBr disk in the range of 400 to 4000 cm^−1^. The morphology and chemical composition of the samples were studied by FE-SEM and EDX analysis using a JEOL JSM-IT 100 (Tokyo, Japan) instrument. The mechanical properties were measured on a universal testing machine (Hounsfield, Redhill, UK). The XRD patterns of the Zn_2_SiO_4_ and SiO_2_–ZnO samples were analyzed on a Bruker XRD D8/Advance/axs instrument (Bruker AXS GmbH, Karlsruhe, Germany).

### 2.2. Preparation of SiO_2_–ZnO Nano-Composite

To a diluted aqueous solution of liquid glass (25 mL) in ethanol (50 mL), ZnCl_2_.4H_2_O (previously dissolved in 50 mL of deionized water) was added drop-wise under magnetic stirring for 30 min and then neutralized using a hydrochloric acid solution (10%). The mixture was slowly combined with aminoethanol (100 mmol) to adjust the pH at 9. The precipitate was filtered, washed with distilled water, and dried and subsequently was calcined at 600 °C in an electric furnace for 2 h. The resultant matter was powdered and analyzed.

### 2.3. Preparation of Zn_2_SiO_4_

To a diluted aqueous solution of liquid glass in ethanol (50 mL), ZnCl_2_.4H_2_O (previously dissolved in 50 mL of deionized water) was added drop-wise under magnetic stirring for 30 min and then neutralized using a hydrochloric acid solution (10%). The ratio of Si/Zn was adjusted to 1:1. The mixture was slowly combined with aminoethanol (100 mmol) to adjust the pH at 9. The precipitate was filtered, washed with distilled water, dried, and subsequently calcined at 1200 °C in an electric furnace for 2 h. The resultant matter was powdered and analyzed.

### 2.4. Sample Preparation

The starting materials including the fillers were mixed with initial ratio percentages depicted in Table 1 and homogenized by an ultrasonic homogenizer. The homogenized mixture was transferred into aluminum alloy molds and vacuumed. Then the light emitting technique using a Coltolux LED (300 mW/cm^2^, Coltene/Whaledent Inc., Cuyahoga Falls, OH, USA) was applied, and samples were cured for 60 s on each side. Each sample was prepared in 3 × 6 mm diameters. The FS results were determined using a universal testing machine (SANTAM Co. STM-400 serial No:818408 (SANTAM ENG. DESIGN, Tehran, Iran)) at a speed of 1 mm/min. The DTS test was conducted according to ADA Standard No. 27 [24]. The FS, FM, and CS measurements were performed with dimensions according to ISO 10477:92 standards [25]. The tests were repeated three times, and with the assistance of Minitab statistical software, the analysis of variance (ANOVA) (*p* < 0.05) was performed. Tukey’s test was used for homogeneous results, and the Dunnett post hoc 3 test was used to analyze the non-homogenous data.

The samples with Zn_2_SiO_4_ were prepared following a similar procedure. 

## 3. Result 

### 3.1. Filler Characterization Results

The phase composition of Zn_2_SiO_4_ and SiO_2_–ZnO nano-materials was primarily recognized through the XRD pattern measurement. The XRD patterns are depicted in Figure 1. The FE-SEM image of the SiO_2_–ZnO nano-composite shows that the particles with a spherical morphology exhibited a particle size distribution of less than 100 nm, while Zn_2_SiO_4_ had larger and more uniform particles (Figure 2). The chemical composition of Zn_2_SiO_4_ and SiO_2_–ZnO was determined by EDX analysis as depicted in Figure 2. Zn_2_SiO_4_ has an atomic ratio of 4/1/2 and a chemical composition of 28.75, 12.49, and 57.35 wt% for O, Si, and Zn elements, respectively. Similarly, SiO_2_–ZnO has an atomic ratio of 3.4/1.2/1 and a chemical composition of 33.02, 20.58, and 45.11 wt% for O, Si, and Zn elements, respectively.

### 3.2. Dental Composite Samples Characterization Results

Ten samples of dental composites filled with 2, 5, 8, 10, and 15 wt% of SiO_2_–ZnO and Zn_2_SiO_4_ were prepared. The samples were coded as **A_1_**–**A_10_** as shown in Table 1. All samples were characterized by different techniques including FT-IR, FE-SEM, EDX, and TGA.

Figure 3 shows the FT-IR analyses of Zn_2_SiO_4_ and SiO_2_–ZnO and dental composite samples. The TGA measurements for all samples were carried out in a nitrogen atmosphere to characterize the thermal stability behaviors of the dental composite samples (Table 1, **A_1_**–**A_10_**). The obtained results are depicted in Figure 4. The morphology and chemical compositions of all samples were investigated by FESEM images and EDX analyses, and the results are shown in Figure 5, Figure 6 and Figure 7. All samples were irregular in shape and composed of C, O, S, and Zn atoms, which confirmed the corrected structure of the dental composites and the presence of inorganic fillers in organic polymer mixtures.

### 3.3. Mechanical Properties Results

All samples were evaluated for their mechanical characteristics including the flexural strength (FS), flexural modulus (FM), diameter tensile strength (DTS), and compressive strength (CS). Figure 8, Figure 9, Figure 10 and Figure 11 show the results of other mechanical parameters including the flexural strength (FS), flexural modulus (FM), diameter tensile strength (DTS), and compressive strength (CS). 

## 4. Discussion

The XRD pattern of SiO_2_–ZnO showed a broad peak below 30 [2θ°] corresponding to the silica (SiO_2_) phase, which seemed to be amorphous (Figure 1). The sharp peaks that appeared after that were around 31.8, 34.4, 36.2, 47.5, 56.6, 62.8, 67.9, and 69.1 [2θ°], which corresponded to the hexagonal structure (JCPDS: 01-079-2205) of zinc oxide (ZnO) with the P63mc space group. The XRD pattern of Zn_2_SiO_4_ showed characteristic peaks at 22.2, 25.4, 31.5, 34.1, 38.8, 47.1, 49.3, 66.0, 70.2, and 77.7 [2θ°] corresponding to a rhombohedral crystal system [Reference code: 00-002-1412] (Figure 1).

The crystal size of the samples was calculated using the Scherrer Equation (1).
(1)$$D=\frac{K\lambda }{\beta \;\mathrm{Cos}\theta }$$
where K is the Scherrer constant, λ is the wavelength of the X-ray beam used (1.54 Å). β is the full-width at half maximum (FWHM) of the peak, and θ is the Bragg angle [19,20,21,22,23]. The main crystal sizes of Zn_2_SiO_4_ and SiO_2_–ZnO samples were calculated as 65 and 59 nm (for the ZnO phase), respectively.

As shown in Figure 3, The FT-IR adsorption spectra of the SiO_2_–ZnO sample show distinctive adsorption bonds at 1618, 1070, 942, and 560 cm^−1^ for stretching and bending vibration of Si–O, Zn–O, and Si–OH bonds. The broad adsorption bond at 3100–3600 cm^−1^ could be referred to OH groups inserted to the surface of SiO_2_ and adsorbed water by KBr discs. Similarly, the Zn_2_SiO_4_ sample has adsorption bands centered at 1626, 1485, 1012, and 510 cm^−1^ for their Si-O, SiO_4_, and Zn–O groups. 

Next, the FT-IR analyses of all samples were performed, and their infrared spectra are depicted in Figure 3. All samples showed a broad adsorption bond from 3200 to 3600 cm^−1^ referred to OH groups and adsorbed water by KBr discs. The adsorption bonds at 3031 (stretching vibration C–H, sp2), 2945, 2870 (stretching vibration C–H, sp3), 1769, 1725 (stretching vibration COO groups), 1630, 1601, 1565 (stretching vibration C=C bonds), 1354 (bending vibration C–H, sp3), 1170, 1135, and 1067 (stretching and bending vibration C–C and C–O bonds) could confirm that organic parts of the composite were present. The filled samples with a lower 10% concentration did not show adsorption bonds related to metal oxide bonds, while in higher dosages, the bonds were broadened at the range 1000–1200 cm^−1^, and a distinctive bond below 600 cm^−1^ confirmed that the inorganic parts of the sample were present.

According to the TGA measurements as shown in Figure 4, the weight losses of all samples had a similar pattern from about 200 to 600 °C due to the thermal decomposition of the organic polymeric networks. Also, a maximum of 2% of weight loss from 50 to 200 °C was due to the adsorbed water and impurities evaporation. The weight loss percents could be matched with the composition of the samples at the first-time construction of the samples (Table 1, **A_1_**–**A_10_**). In addition, the filled samples showed higher thermal stability than those of the unfilled ones.

As can be seen from Figure 8, the FS of the samples was increased when the concentration of the filler was increased, and the highest increase was observed in 8 wt% (**A_3_**, 123.4 MPa) of SiO_2_–ZnO (*p* > 0.05). This value for Zn_2_SiO_4_ as a filler was 10 wt% (**A_9_**, 136.6 MPa). With the further increase in filler dosages, the FS values decreased. 

Generally, compared with the dental resin base sample with a compressive strength value of 97.6 MPa, the samples containing SiO_2_–ZnO as a filler with 2, 5, 8, 10, and 15 wt% showed higher strength. Zn_2_SiO_4_ as a filler showed higher values of FS than SiO_2_–ZnO. 

A similar trend was found when other mechanical parameters including FM, DTS, and CS were investigated. Notably, with the increase in filler content, an increase in the FM, DTS, and CS characteristics was observed. Accordingly, 8 wt% of fillers (samples SiO_2_–ZnO (**A_3_**) and Zn_2_SiO_4_ (**A_8_**)) showed higher values of FM, DTS, and CS characteristics as 8.6 GPa, 34.2 MPa, and 183.8 MPa values for SiO_2_–ZnO and 11.3 GPa, 41.2 MPa, and 190.5 MPa values for Zn_2_SiO_4_ samples (Figure 9, Figure 10 and Figure 11). The improvement of the mechanical properties of dental resins was due to the adhesion properties between the resins and fillers. In the meantime, due to the characteristic of high polarity, Zn_2_SiO_4_ created the possibility of a more suitable connection and more delay compared with SiO_2_–ZnO. This characteristic showed itself in the higher values of the measured FS, FM, DTS, and CS characteristics. The acceptable values for the DTS test of a dental composite were 30–55 MPa [26], which was consistent with the results of SiO_2_–ZnO and Zn_2_SiO_4_ in the current study (samples **A_3_** (34.2 MPa) and A8 (41.2 MPa)). In all cases, the *p* value was significant (*p* ˂ 0.05).

In comparison, SiO_2_ and ZnO as fillers alone or together did not improve the mechanical properties as well as SiO_2_–ZnO or Zn_2_SiO_4_ (Table 2). The results showed that the filler dosages and types significantly influenced the mechanical characteristics of the dental resins. On the other hand, Zn_2_SiO_4_ as a filler performed better results than the SiO_2_–ZnO sample. A historical view of point revealed that a glass (Specialty Glass, USA, 2–4m, 73 wt%) was used by Zandinejad et al. for reinforcing and improving the mechanical characteristics of bis-GMA/TEGDMA (70/30) mixed resins and showed suitable improvement in the FS, FM, and DTS mechanical properties [27]. Samuel and co-workers reported that mesoporous and nonporous spherical silica (70 wt%) could improve the mechanical properties of dental resins [28]. In another study, silica nanoparticle loading showed suitable increasing in the FS in bis-GMA/TEGDMA mixed resins [29]. However, when compared, the results of the SiO_2_–ZnO and Zn_2_SiO_4_ samples were comparable and, in some cases, better than those of the fillers reported in the literature (Table 2).

This work suggested that Zn_2_SiO_4_ and SiO_2_–ZnO could effectively improve the thermal stability and mechanical properties of Bis-GMA/ TEGDMA dental resins when used as reinforcing filler. In comparison with the fillers reported in the literature, Zn_2_SiO_4_ and SiO_2_–ZnO had reasonably good potential to improve the FS, FM, DTS, and CS characteristics of dental resins. Worth noting about ZnO and its compounds is their bioactivity, which may increase the applications of this work. For future research, Zn_2_SiO_4_ and SiO_2_–ZnO could be used for the improvement of different dental resins and polymers. In addition, experiments related to biological tests could be the next study.

The main limitation of this work is the use of only Bis-GMA/TEGDMA dental resins, and also, this work did not investigate the dental tests. Also, the cytotoxicity and biological effects of these fillers were not studied. Thus, this work could be expanded by other researchers for dental applications. In addition, the studies with CAD/CAM blocks and color stability of the samples are of interest for future investigations [33].

## 5. Conclusions

Zn_2_SiO_4_ and SiO_2_–ZnO samples were prepared and characterized by XRD, SEM, EDX, and FT-IR techniques. The samples were used as a filler in a dental composite constructed from Bis-GMA and TEGDMA copolymers. Compared with non-composited dental resins, a dental composite containing Zn_2_SiO_4_ and SiO_2_–ZnO as a filler had superior values for FS, FM, DTS, and CS characteristics. As found, 8 wt% of SiO_2_–ZnO and 10 wt% of Zn_2_SiO_4_ showed higher values of FS of 123.4 and 136.6 MPa, respectively. The optimized values of the FM, DTS, and CS parameters for Zn_2_SiO_4_ (8 wt%) as a filler were 131.3 MPa, 11.33 GPa, 41.25 MPa, and 190.56 MPa, and those for SiO_2_–ZnO (8 wt%) were 123.4 MPa, 8.64 GPa, 34.24 MPa, and 183.83 MPa, respectively. To date, various fillers have been used to improve the mechanical properties of dental resins. Among the different fillers, nanocomposites seem to provide promising results for enhanced mechanical properties. Silica nanocomposites are inexpensive and show reasonable improvements regarding the mechanical properties of various polymeric systems. Thus, the use of silica–metal oxide nanocomposites could definitely improve the mechanical properties of dental composites.

## Figures and Tables

**Figure 1 materials-16-06268-f001:**
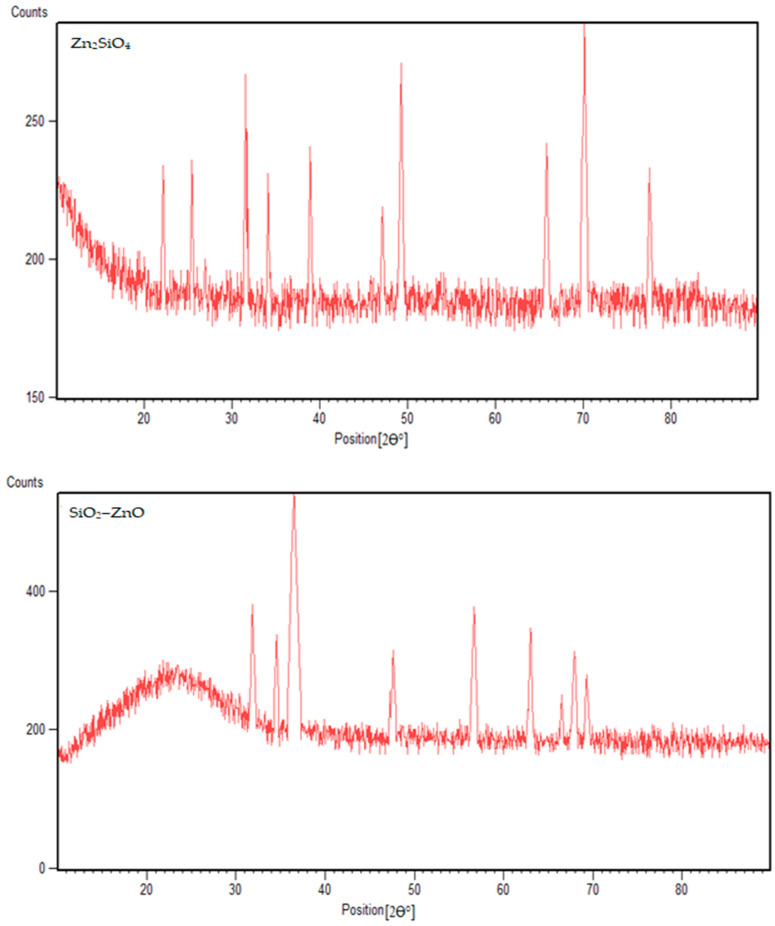
XRD patterns of Zn_2_SiO_4_ and SiO_2_–ZnO.

**Figure 2 materials-16-06268-f002:**
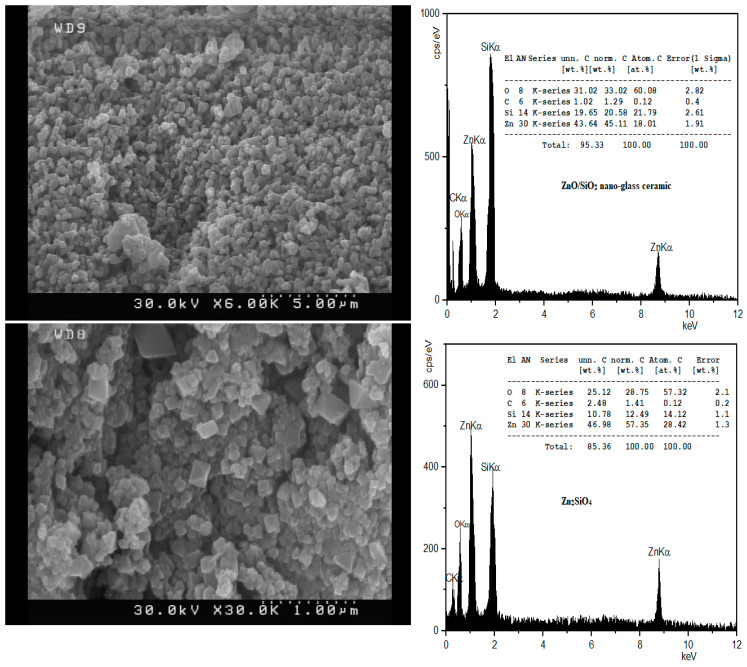
FE-SEM images and EDX analyses of Zn_2_SiO_4_ and SiO_2_–ZnO.

**Figure 3 materials-16-06268-f003:**
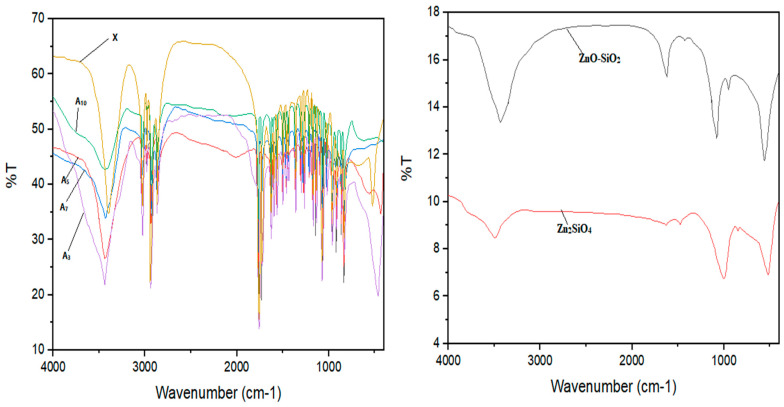
FT-IR analyses of Zn_2_SiO_4_ and SiO_2_–ZnO and dental composite samples. The left image illustrates the FT-IR analyses of dental samples (**A_3_**–**A_5_**–**A_7_**–**A_10_**, and X), The right image shows FT-IR analyses of Zn_2_SiO_4_ and ZnO-SiO_2_.

**Figure 4 materials-16-06268-f004:**
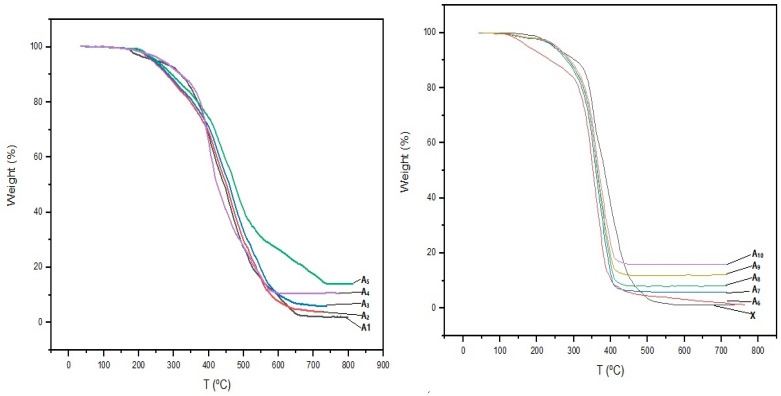
TGA analyses of dental composite samples (**A_1_**–**A_10_**) filled with Zn_2_SiO_4_ and SiO_2_–ZnO. In accordance with Table 1, the image on the left displays TGA analyses (**A_1_**–**A_5_**) conducted on dental composite samples, while the image on the right features samples (**A_6_**–**A_10_** and X).

**Figure 5 materials-16-06268-f005:**
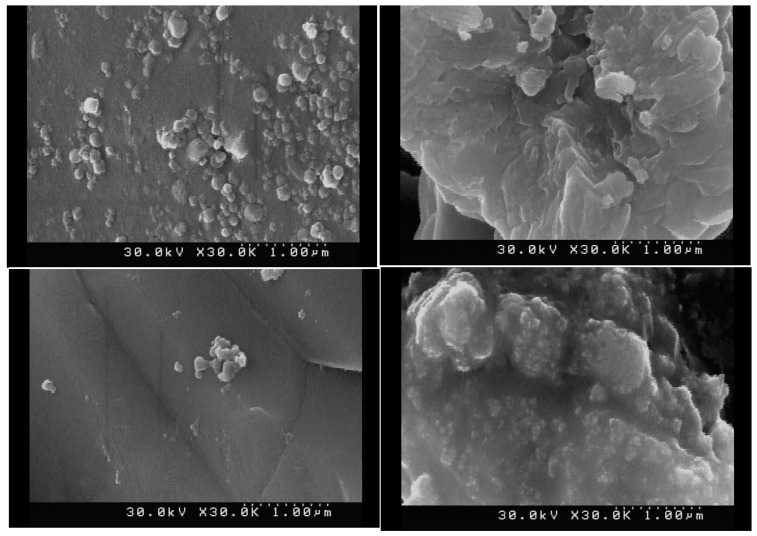
Selected FE-SEM images of dental composite samples (**A_1_**–**A_10_**).

**Figure 6 materials-16-06268-f006:**
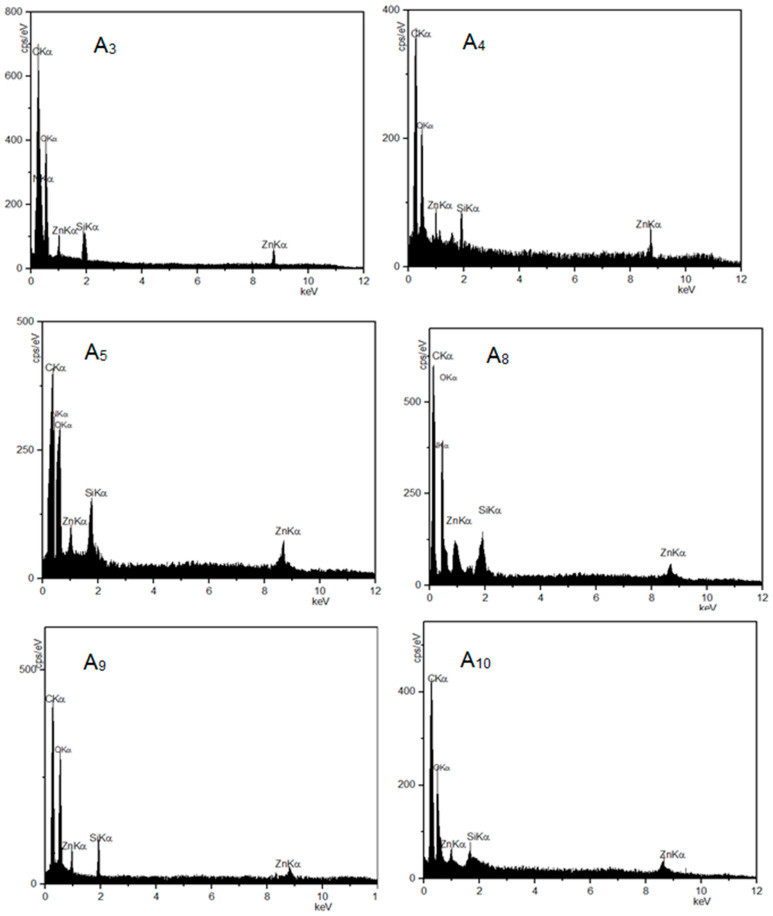
FE-SEM and EDX analyses of dental composite samples (**A_3_**, **A_4_**, **A_5_**, **A_8_**, **A_9_**, and **A_10_**).

**Figure 7 materials-16-06268-f007:**
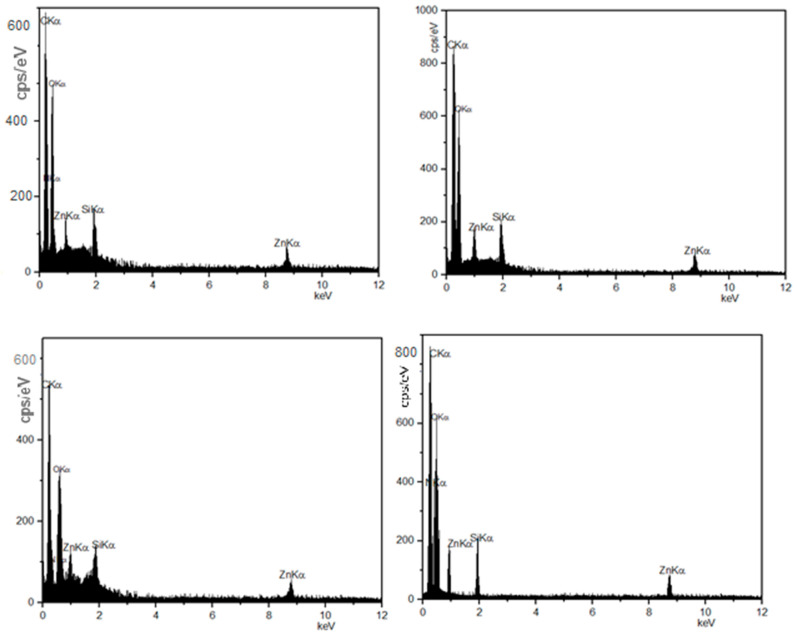
FE-SEM and EDX analyses of dental composite samples (**A_1_**, **A_2_**, **A_6_**, and **A_7_**).

**Figure 8 materials-16-06268-f008:**
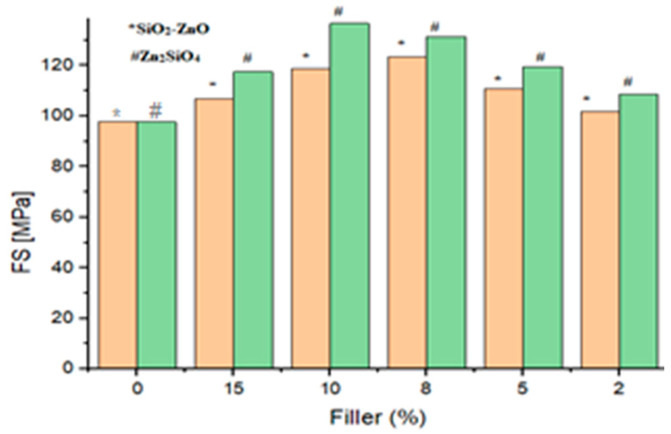
Flexural strength (FS) of dental composites (**A_1_**–**A_10_**).

**Figure 9 materials-16-06268-f009:**
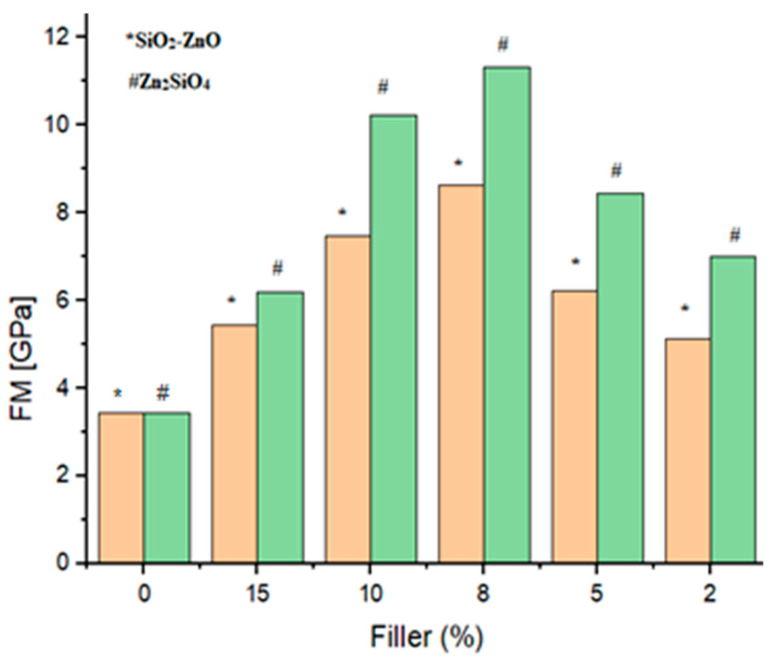
Flexural modulus (FM) of dental composites (**A_1_**–**A_10_**).

**Figure 10 materials-16-06268-f010:**
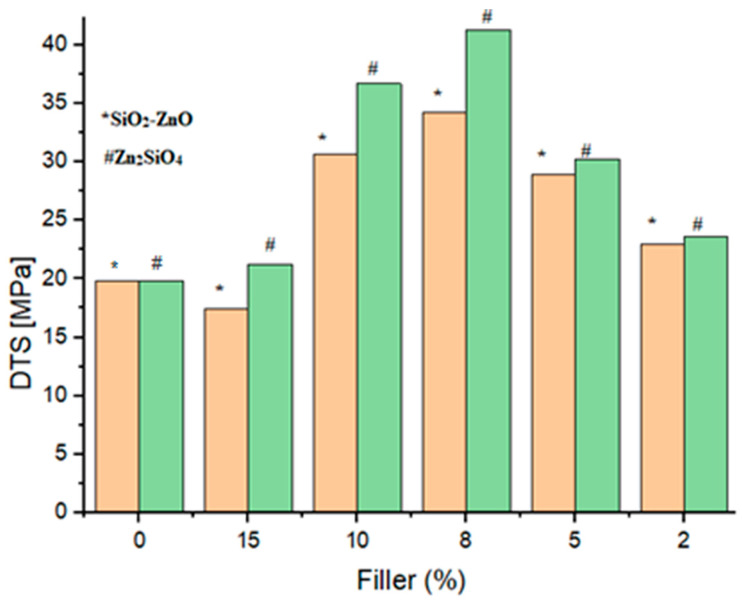
Flexural diameter tensile strength (DTS) of dental composites (**A_1_**–**A_10_**).

**Figure 11 materials-16-06268-f011:**
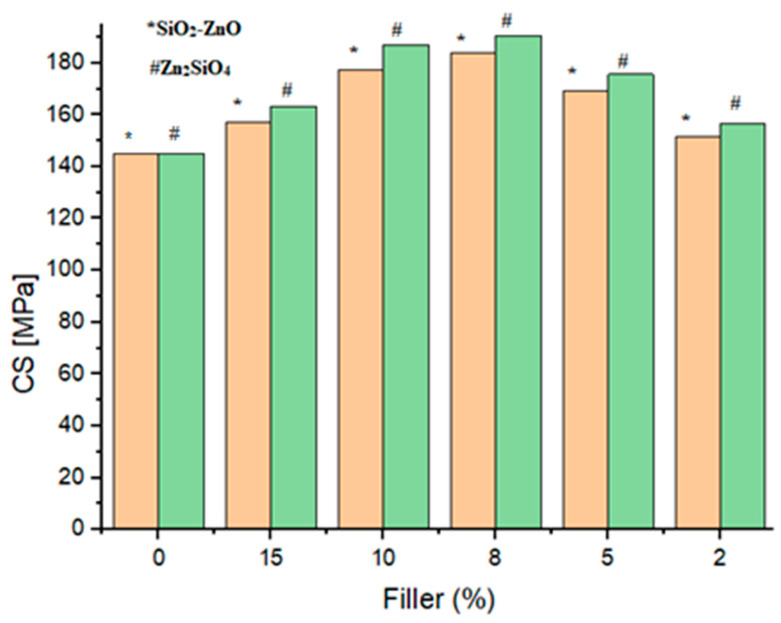
Compressive strength (CS) of dental composites (**A_1_**–**A_10_**).

**Table 1 materials-16-06268-t001:** Coded samples of dental composites: the starting materials were mixed according to the wt% of each material.

Sample Code	Bis-GMA (wt%)	TEGDMA (wt%)	CQ/3-NDADM (wt%)	Filler (wt%)
**X**	70	29	0.5/0.5	-
**A_1_**	69	28	0.5/0.5	SiO_2_–ZnO (2)
**A_2_**	67.5	26.5	0.5/0.5	SiO_2_–ZnO (5)
**A_3_**	66	25	0.5/0.5	SiO_2_–ZnO (8)
**A_4_**	65	24	0.5/0.5	SiO_2_–ZnO (10)
**A_5_**	62.5	21.5	0.5/0.5	SiO_2_–ZnO (15)
**A_6_**	69	28	0.5/0.5	Zn_2_SiO_4_ (2)
**A_7_**	67.5	26.5	0.5/0.5	Zn_2_SiO_4_ (5)
**A_8_**	66	25	0.5/0.5	Zn_2_SiO_4_ (8)
**A_9_**	65	24	0.5/0.5	Zn_2_SiO_4_ (10)
**A_10_**	62.5	21.5	0.5/0.5	Zn_2_SiO_4_ (15)

**Table 2 materials-16-06268-t002:** Comparison results.

Filler	Bis-GMA/TEGDMA wt%	FS [MPa]	FM [GPa]	DTS [MPa]	CS [MPa]	Ref.
-	70/29.45	97.9	3.4	19.8	144.9	This work
**A_3_**	65/26.45	123.4	8.64	34.24	183.83	This work
**A_8_**	65/26.45	131.3	11.33	41.25	190.56	This work
SiO_2_–ZnO bulk (8 wt%)	65/24.45	105.74	6.45	24.14	155.31	This work
SiO_2_ (6wt%) + ZnO (2 wt%)	65/24.45	115.02	7.82	31.21	177.24	This work
ZnO (6 wt%)	68/29.45	91.25	3.26	17.09	142.02	This work
SiO_2_ (6 wt%)	68/29.45	103.49	4.9	26.13	155.67	This work
Glass (Specialty Glass, Willow Grove, PA, USA) 2–4 m. (73 wt%)	70/30	43.6	8.6	42.5	-	Zandinejad et al. [27]
Mesoporous and nonporous spherical silica (70 wt%)	50/50	72	6.7	-	191	Samuel et al. [28]
Silica nanoparticles modified with γ-methacryloxy propyl trimethoxy silane (40 wt%)	70/30	149.74	-	-	-	Hosseinalipour et al. [29]
Quaternary ammonium polyethylenimine nanoparticles (2 wt%)	60/40	63.09	3.712	-	-	Barszczewska-Rybarek et al. [5]
Short glass fibers (70 wt%)	70/30	-	4.59	-	66.6	Krause et al. [30]
Urchin-like hydroxyapatite (10 wt%)	49.5/49.5	123.5	-	42.7	363.5	Liu et al. [31]
Electrospun nylon 6 nanofiber (7.5 wt%)	50/50	112.1	2.726	-	-	Fong [32]

## Data Availability

Data will be made available on request.

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
