# Peer review of "Glass-Ceramic Fillers Based on Zinc Oxide–Silica Systems for Dental Composite Resins: Effect on Mechanical Properties"

_materials, 2023, doi:10.3390/ma16186268_

Round 1

Reviewer 1 Report

Dear Authors;

In this manuscript, a new glass-ceramic system to be used as a filler for dental composite reinforcements were developed. In continue of these studies on the preparation and application of nanostructured materials, Zn2SiO4 and SiO2-ZnO nano-glass ceramics were used as resistant reinforcing particles in a dental composite to measure mechanical properties such as flexural strength and etc. after construction.

An important topic was selected within the scope of the journal. It is a useful addition to the literature on dentistry applications. In general, the paper is well written, and the results were articulated and compared. The manuscript can be considered with minor correction as below: 

  1. Why did you choose this study?  Please provide further justification.
  2.  It is necessary to indicate the originality of the current work.
  3. The abstract seems to be a little underdeveloped. It would be better to reword it and include a slightly more detailed explanation of the study. This can help the readers have a better and quick understanding of the study.
  4. The analysis and evaluation section should report the findings, interpretation, and detailed discussion (So, this important section needs to be expanded).
  5. Does your work add something new compared to previous studies? (Explain this point in discussion).
  6. It will be good to add the limitations of this study in the discussion
  7.  The recommendations for future work might be considered.

Thank you

Some sentences are grammatically wrong. Before the submission of the revised manuscript, the English language of your manuscript must be professionally proofread and edited.

Reviewer 2 Report

The article “Zn2SiO4 and SiO2-ZnO nano-glasses fillers: efficient promoters for the improvement of the mechanical properties of experimental resin dental composites” aims to to develop a new glass-ceramic system to be used as a filler for dental composite reinforcements. 

The article cover an interesting topic, nevertheless some major improvements are to be performed before possible publication.

The text:

“Revisiting the research conducted in the field of optimizing the properties of dental polymers, there are many interesting reports. Polyacrylonitrile (PAN) nanofibers is used by Amiri 48 et-al. for reinforcing and improving of bis-GMA/TEGDMA mixed resins. The results show suitable improvement on the FS, FM, and CS mechanical properties [21]. Zeng and coworkers reported that carbon nanotubes could improve the mechanical properties of dental resins when modified by silane coupling agents [22]. In another research, Ag nanoparticles loading showed suitable increasing on the FS, FM, water sorption and hardness of bis-GMA/TEGDMA mixed resins [23]. In other words, “

is related to what has been tested in the literature during the past years. The reviewer thinks that this text should be removed or moved to the discussion.

A list of the multiple type of fillers is enough (as correctly listed from lines 54 to 60.

Line 77: from Merck company: please add City, State and Country of Merck company

Add Manufacturer, City, State and Country for every material or device used in the paper.

ZnCl2.4H2O Use Superscript or subscript in the formulas.

The Discussion is mixed with the results, the reviewer thinks it should be splitted.

The article needs to compare the results with similar articles in the Discussion. If no similar articles (same fillers, same materials) are found, compare them with other types of fillers to highlight the advantages or disadvantages of the tested ones.

Can these fillers be used on CAD/CAM polymer (composite) blocks?

The authors did not mention possible consequences on color stability or translucency of the investigated fillers.

Will this have a consequence on color stability? Consider citing https://doi.org/10.3390/polym15020464

What are the limitations of the study

Which possible future developments/studies are to be performed considering the current paper's conclusions?

Reviewer 3 Report

Dear Author,

This important work for dental applications to improve mechanical properties of dental implantable materials.

However i have following comments 

1.Introduction important results need to discuess quantitatively and statistically?

2.Reagents need to write with analytical grade and country of supplier so it will be easier for other to easily reproduce it?

3.Figure 1. XRD patterns of Zn2SiO4 and SiO2-ZnO nano-glass ceramic Theta should be written with symbols ÆŸ ?

4.Figure 4. TGA analyses of dental composite samples (A1-A10) filled with Zn2SiO4 and SiO2-ZnO 189 nano-glass ceramic all parameters from this Figure need to add in separate table like residue remain,half decomposition,initial decompostion and other parameters?

5.Conclusion should be answer to your hopothesis or Novelty followed by discuesssion?

NA 

Reviewer 4 Report

This manuscript is talking about the influence of Zn2SiO4 and SiO2-ZnO nano-glass ceramics on the mechanical properties of Bis-GMA/TEGDMA mixed dental resins. Since the idea is focusing on enhancing the mechanical properties of the dental material, it would be better if authors would like to present more mechanical characterizations, such as micro-hardness and elastic modulus. Besides, the introduction will be more well-round, if authors would like to add more contents identifying the research gap

Minor language improvement 

Reviewer 5 Report

This paper is a study of specific fillers in experimental dental composite resin systems.  As such, it is one of numerous studies of this type, and in its current form adds very little to our state of knowledge.  The paper must by substantially revised before it can be considered for publication.

Here are my detailed comments:

The title is poorly written, and the English needs attention.  A better version would be "Glass-ceramic fillers based on zinc oxide-silica systems for dental composite resins: Effect on mechanical properties."

Line 33: Fillers have no effect on biocompatibility.  Delete this mention.

Line 34: There are not three phases in dental composites, there are two: polymer phase + filler.  The initiators are part of the polymer phase. It is arguable that there are three types of component, monomer, initiator and filler.  In fact there are more, including coupling agent to bond the filler to the polymer phase.

Line 39: The chemical bisphenylglycidyl dimethacrylate should not begin with a capital letter.

Line 44: You do not need 20 references to support this statement.  Of those cited, numbers 8, 9, 10 and 19 are especially inappropriate, since they do not deal with mechanical properties. In my opinion, all 20 could be replaced with a single reference, number 14.  It is a review and covers every aspect of the selection and effect of fillers in these composite materials.

Line 67: "Biocompatibility" is a meaningless term without mentioning the biological context.  I recommend changing this to "... biocompatibility with the tooth."

Line 71: English is especially poor here.

Also, what is the difference between the materials?  They are both based on ZnO and SiO2, and nowhere is it explained why the authors keep listing the materials as two types.  Finally, what is a "nano-glass ceramic".  If this term actually means anything distinct, it should be explained in the Introduction.

Line 166: There is a strange statement here.  Surely it is obvious that there are organic components here!  It did not need FTIR to demonstrate the fact.

Line 195: Do not use the term "synthesized". It is not appropriate.  "Synthesized" is a word properly applied to making new molecules.  Blending existing substances together should be described as "fabricated" or "formulated".

Lines 197 and 208: Units should be MPa, not Mpa.

Line 212: The authors state that Zn2SiO4 creates "a more suitable connection and more delay." Compared with what? Why is a delay desirable? How is this filler better than a conventional filler such as quartz?  Both materials are inorganic powders, and surely they both have similar polarities?  This attempted discussion is all most confusing, probably because the authors themselves are obviously confused.

Line 274.  It is ridiculous to compare the mechanical properties of the experimental composite with the unfilled polymer system, as the authors appear to be doing.  It is more to the point to compare the properties with a composite filled with a conventional filler, such as quartz, at the same volume loading.

Overall, this is a very disappointing paper, and thorough revision is essential.

English is in need of considerable attention, as indicated.

Round 2

Reviewer 2 Report

All the reviewer's requests have been amended

Reviewer 3 Report

Dear Author,

I appreciated efforts of author for improving scientific and technical content of paper.

Reviewer 4 Report

Approve for publication

No need

Reviewer 5 Report

The authors have undertaken a substantial revision of the original manuscript and have addressed all of the concerns I had.  I therefore consider the paper has now been revised sufficiently for it to go forward to publication.